# Neoadjuvant Treatment with HER2-Targeted Therapies in HER2-Positive Breast Cancer: A Systematic Review and Network Meta-Analysis

**DOI:** 10.3390/cancers14030523

**Published:** 2022-01-21

**Authors:** Agampodi Danushi M. Gunasekara, Thunyarat Anothaisintawee, Sitaporn Youngkong, Nguyen T. Ha, Gareth J. McKay, John Attia, Ammarin Thakkinstian

**Affiliations:** 1Mahidol University Health Technology Assessment Graduate Program (MUHTA), Mahidol University, Bangkok 10400, Thailand; danushigunasekara@kdu.ac.lk (A.D.M.G.); nthiha@medvnu.edu.vn (N.T.H.); ammarin.tha@mahidol.edu (A.T.); 2Department of Paraclinical Sciences, Faculty of Medicine, General Sir John Kotelawala Defence University, Ratmalana, Colombo 10390, Sri Lanka; 3Department of Family Medicine, Faculty of Medicine, Ramathibodi Hospital, Mahidol University, Bangkok 10400, Thailand; 4Department of Clinical Epidemiology and Biostatistics, Faculty of Medicine, Ramathibodi Hospital, Mahidol University, Bangkok 10400, Thailand; 5Social and Administrative Pharmacy Division, Department of Pharmacy, Faculty of Pharmacy, Mahidol University, Bangkok 10400, Thailand; 6School of Medicine, Vietnam National University, Ho Chi Minh City 700000, Vietnam; 7Centre for Public Health, School of Medicine, Dentistry and Biomedical Sciences, Queen’s University, Belfast BT12 6BA, UK; g.j.mckay@qub.ac.uk; 8School of Medicine and Public Health, College of Health and Wellbeing, University of Newcastle, Newcastle, NSW 2308, Australia; john.attia@newcastle.edu.au

**Keywords:** targeted therapy, HER2-positive breast cancer, neoadjuvant therapy, systematic review, network meta-analysis

## Abstract

**Simple Summary:**

Human epidermal growth factor receptor 2 (HER2)-positive breast cancer causes more aggressive progression of disease and poorer outcomes for patients. HER2-targeted medicines used as neoadjuvant systemic therapy could improve clinical outcomes in early-stage or locally advanced breast cancer patients. The purpose of this systematic review and network meta-analysis was to identify the neoadjuvant anti-HER2 therapy with the best balance between efficacy and safety. We found that trastuzumab emtansine + pertuzumab + chemotherapy had a high pathologic complete response with a low risk of adverse events compared to other neoadjuvant anti-HER2 regimens, while the pertuzumab + trastuzumab + chemotherapy regimen showed the highest disease-free survival. However, further trial data on neoadjuvant regimens with trastuzumab emtansine are needed to confirm these findings.

**Abstract:**

This systematic review aimed to identify neoadjuvant anti-human epidermal growth factor receptor 2 (HER2) therapies with the best balance between efficacy and safety. Methods: A network meta-analysis was applied to estimate the risk ratios along with 95% confidence intervals (CIs) for pathological complete response (pCR) and serious adverse events (SAE). A mixed-effect parametric survival analysis was conducted to assess the disease-free survival (DFS) between treatments. Results: Twenty-one RCTs with eleven regimens of neoadjuvant anti-HER2 therapy (i.e., trastuzumab + chemotherapy (TC), lapatinib + chemotherapy (LC), pertuzumab + chemotherapy (PC), pertuzumab + trastuzumab (PT), trastuzumab emtansine + pertuzumab (T-DM1P), pertuzumab + trastuzumab + chemotherapy (PTC), lapatinib + trastuzumab + chemotherapy (LTC), trastuzumab emtansine + lapatinib + chemotherapy (T-DM1LC), trastuzumab emtansine + pertuzumab + chemotherapy(T-DM1PC), PTC followed by T-DM1P (PTC_T-DM1P), and trastuzumab emtansine (T-DM1)) and chemotherapy alone were included. When compared to TC, only PTC had a significantly higher DFS with a hazard ratio (95% CI) of 0.54 (0.32–0.91). The surface under the cumulative ranking curve (SUCRA) suggested that T-DM1LC (91.9%) was ranked first in achieving pCR, followed by the PTC_T-DM1P (90.5%), PTC (74.8%), and T-DM1PC (73.5%) regimens. For SAEs, LTC, LC, and T-DM1LC presented with the highest risks (SUCRA = 10.7%, 16.8%, and 20.8%), while PT (99.2%), T-DM1P (88%), and T-DM1 (83.9%) were the safest regimens. The T-DM1PC (73.5% vs. 71.6%), T-DM1 (70.5% vs. 83.9%), and PTC_T-DM1P (90.5% vs. 47.3%) regimens offered the optimal balance between pCR and SAE. Conclusions: The T-DM1PC, T-DM1, and PTC_T-DM1P regimens had the optimal balance between efficacy and safety, while DFS was highest for the PTC regimen. However, these results were based on a small number of studies, and additional RCTs assessing the efficacy of regimens with T-DM1 are still needed to confirm these findings.

## 1. Introduction

Breast cancer is the most common cancer among women worldwide, with 2.26 million new cases in 2020. It ranked fifth as the leading cause of mortality and accounted for 15% of cancer deaths in women [1]. Patients with human epidermal growth factor receptor 2 (HER2) positivity, accounting for approximately 20 to 24% of breast cancer patients globally, had more aggressive progression and poorer outcomes than those who were HER2-negative. There have been many recent advances in HER2-targeted therapies, including trastuzumab (T), lapatinib (L), pertuzumab (P), and trastuzumab emtansine (T-DM1), which have significantly improved the progression and overall survival outcomes for HER2+ve breast cancer patients [2,3].

In addition, HER2-targeted therapies have been used as neoadjuvant systemic therapy (i.e., regimen started before surgery) with the aim of improving operability and the pathological complete response (pCR) in early-stage or locally advanced breast cancers [3,4,5]. Many regimens, including (1) a combination of single HER2-targeted agent with chemotherapy (C) (i.e., trastuzumab + chemotherapy (TC), pertuzumab + chemotherapy (PC), lapatinib + chemotherapy (LC)), (2) dual HER2-targeted agents with chemotherapy (i.e., pertuzumab + trastuzumab + chemotherapy (PTC), lapatinib + trastuzumab +chemotherapy(LTC)], or (3) dual HER2-targeted agents without chemotherapy (i.e., pertuzumab + trastuzumab (PT), trastuzumab emtansine + pertuzumab (T-DM1P)) have been used as neoadjuvant systemic therapies for early-stage or locally advanced breast cancers. The European Society for Medical Oncology (ESMO) and the National Comprehensive Cancer Network (NCCN) guidelines have recommended trastuzumab as standard neoadjuvant therapy for HER2+ve breast cancer and trastuzumab combined with pertuzumab for high-risk patients with node positivity and/or estrogen receptor negativity [6,7]. However, given the availability of several regimens with limited head-to-head comparisons from randomized controlled trials (RCTs), the identification of the optimal regimen of neoadjuvant with HER2-targeted therapy for HER2+ve breast cancers remains challenging.

Network meta-analysis (NMA) is a method that combines data from direct comparisons to indirectly compare between different interventions by borrowing information from common comparators. In addition, NMA can estimate the probability associated with being the optimal treatment and enables the ranking of both positive and negative effects. Two NMAs published in 2018 and 2019 [8,9] identified the PTC regimen as the best treatment for pCR, whereas T-DM1P was the safest regimen with the lowest probability associated with adverse events. More recently, several RCTs have assessed newer neoadjuvant anti-HER2 regimens (i.e., T-DM1 alone, T-DM1 + lapatnib + chemotherapy (T-DM1LC), T-DM1 + pertuzumab + chemotherapy (T-DM1PC), and PTC +T-DM1P combination regimens) [10,11]. Furthermore, additional patient-relevant outcomes, such as disease-free survival (DFS), were not previously considered in the prior NMAs due to insufficient data. Therefore, this systematic review and NMA aimed to identify the current regimens of neoadjuvant anti-HER2 therapy with the highest probability of DFS and pCR coupled with the lowest risk of adverse events.

## 2. Materials and Methods

This systematic review was reported according to the Preferred Reporting Items for Systematic Reviews and Meta-Analyses (PRISMA) guidelines, and the review protocol was registered at the PROSPERO website (CRD42020211532).

### 2.1. Literature Search and Selection of Studies

Relevant studies were searched in Medline via PubMed, Scopus, and Cochrane Central Register of Controlled Trials from their inceptions to November 2021. Details of search terms and search strategies for each database are provided in Appendix B (see Table A1, Table A2 and Table A3).

#### 2.1.1. Selection of Studies

Studies were selected by two independent reviewers (A.D.M.G. and N.T.H.). Randomized controlled trials were eligible if they (1) included early-stage and/or locally advanced HER2+ve breast cancer patients, (2) compared any neoadjuvant regimens that included anti-HER2 therapies, with or without chemotherapy, and (3) reported any of the following outcomes: pCR, DFS, or overall survival. Trials that included HER2+ve breast cancer with distant metastasis or compared the different doses of the same treatment regimen were excluded.

#### 2.1.2. Interventions of Interest

Interventions of interest included neoadjuvant therapies of HER2-targeted agents with or without chemotherapy, such as: Single HER2-targeted agents with chemotherapy (i.e., TC, PC, LC);Dual HER2-targeted agents with chemotherapy (i.e., PTC, LTC, T-DM1PC, T-DM1LC, PTC_T-DM1P). T-DM1PC was a response guided regimen where T-DM1P was provided in cycles 1 through 4, followed by 4 cycles of chemotherapy in non-responders or a continuation of 2 cycles of T-DM1P in responders. PTC_T-DM1P was a regimen where PTC was provided in cycles 1 through 4, followed by T-DM1P in cycles 5 through 8;Dual HER2-targeted agents without chemotherapy (i.e., PT, T-DM1P);Single HER2-targeted agents without chemotherapy (i.e., T-DM1);

Comparators were chemotherapy alone or different regimens of neoadjuvant anti-HER2 agents.

#### 2.1.3. Outcomes of Interest

Primary outcomes of interest were DFS, overall survival, and pCR. DFS was defined as the time from randomization to first locoregional recurrence, contralateral breast cancer or distant metastasis or death from any causes [12]. Although overall survival was defined as the time from randomization to death from any cause [12,13], the limited number of studies reporting this outcome prevented its evaluation. Pathological complete response was defined as no histological evidence of residual invasive tumor cells in the breast and axillary lymph nodes (ypT0/Tis and ypN0) [5,14].

The secondary outcomes were (1) serious adverse events (SAE), which included grades 3–4 adverse events (i.e., life threatening events, events requiring hospitalization) according to National Cancer Institute Common Terminology Criteria (NCICTC) [15], and (2) breast conservation surgery (BCS) as the final surgical procedure.

### 2.2. Data Extraction

Data were extracted by two independent reviewers (A.D.M.G., N.T.H.), including study characteristics (i.e., setting, follow-up time, participant numbers), patient characteristics (i.e., mean age, cancer stage, tumor size, nodal status, hormone receptor status, body mass index (BMI), family history of breast cancer, breast feeding, menopausal status), intervention characteristics (i.e., treatment regimens, dosages, course), and types of outcomes. For data pooling, contingency data between treatments and outcomes were extracted. For DFS, time (on *x*-axis), probability of outcome occurrence (on *y*-axis), number of events, and person-time at risk at each time point were extracted from Kaplan–Meier (KM) curves using webplot digitizer software [16].

### 2.3. Risk of Bias Assessment

Risk of bias of each study was assessed using the revised Cochrane risk-of-bias tool (RoB 2) by one reviewer (A.D.M.G.) and randomly checked by senior author (T.A.) [17,18]. Five domains (i.e., randomization, protocol deviation, missing outcome data, measurement of outcome, selection of results reported) were assessed and ranked as low risk, some concern, and high risk. An overall ROB was further classified as low, some concern, and high risk of bias; if all five domains were ranked as low risk, at least one domain was ranked as some concern without high risk, and at least one domain was ranked as high risk, respectively.

### 2.4. Statistical Analysis

Direct meta-analysis was performed if two or more studies with similar treatment comparisons and outcomes were available. Risk ratios (RR) of pCR, BCS, and SAEs were estimated and pooled across studies using the inverse variance method (fixed-effects) if there was no heterogeneity; otherwise, the DerSimonian–Laird method (random effects) was applied. Heterogeneity was assessed using Cochran’s Q test and I^2^ statistic. Heterogeneity was present if the *p*-value from Q-test < 0.10 and/or I^2^ > 25%. Sources of heterogeneity were explored using a meta-regression by fitting co-variables (i.e., node positivity, hormone receptor negativity, chemotherapy regimens (i.e., taxane and taxane + anthracycline) and duration of anti HER2 treatment) one by one. Further subgroup analysis was performed if the Tau^2^ value decreased by more than 50% after fitting the co-variable in the meta-regression model. Publication bias was assessed using Egger’s tests and funnel plots. If there was asymmetry of the funnel plot, a contour-enhanced funnel plot was generated to identify the cause of asymmetry (i.e., small study effect or heterogeneity) [19,20].

The NMAs for pCR, SAE, and BCS were performed using a two-stage NMA approach. First, a relative treatment effect (lnRR) was estimated along with the variance–covariance matrix using a binary regression model. Second, a multivariate meta-analysis with consistency model was applied to pool the lnRRs across studies. Treatment ranking was done using the rankogram and the surface under cumulative ranking curve (SUCRA). Cluster ranking considering both efficacy (pCR) and safety (SAE) was also performed to simultaneously weigh the risks and benefits of each intervention [21,22].

For DFS, the number of events, patients at risk, and probability of DFS at each time point were used to simulate individual patient data IPD for each study [23], and then pooled across all studies. A mixed-effect parametric survival analysis with Weibull distribution was applied to estimate relative treatment effects, i.e., hazard ratio (HR) between different neoadjuvant HER2-targeted regimens [24,25,26,27].

A comparison-adjusted funnel plot was constructed to assess publication bias across the network. The consistency assumption was assessed using design-by-treatment interaction inconsistency model [28]. If there was inconsistency (*p*-value < 0.05), treatment loops having high inconsistency factors (IF) were explored. Subgroup analysis was then performed by excluding studies with different characteristics to improve consistency of the model. All statistical analyses were conducted using STATA program version 16.0. A *p*-value less than 0.05 was considered statistically significant for all tests except for Cochran’s Q test, for which a *p*-value less than 0.10 was applied.

## 3. Results

Of the 3010 studies identified, 32 published studies [5,10,11,12,29,30,31,32,33,34,35,36,37,38,39,40,41,42,43,44,45,46,47,48,49,50,51,52,53,54,55,56], including 21 RCTs [10,11,29,31,32,34,35,38,39,40,43,44,45,46,47,48,49,50,51,53,54], were eligible for inclusion within this analysis (Figure 1). Among these, four anti-HER2 medicines (i.e., T, L, P, T-DM1) were evaluated in 11 regimens (i.e., TC, LC, PC, PT, T-DM1P, PTC, LTC, T-DM1LC, T-DM1PC, PTC_T-DM1P, T-DM1) along with chemotherapy alone (see Appendix A). The individual study characteristics are presented in Table 1. Twelve RCTs were comprised of two arms, for which TC vs. C was the most common treatment comparison (5 RCTs) [30,32,34,35,45], followed by TC vs. LC (2 RCTs) [12,46], with the remainder comprising unique comparisons [10,49,50,51,54]. Eight RCTs included three arms with seven RCTs comparing TC vs. LC vs. LTC [31,40,43,44,47,48,53]; a single RCT compared PTC vs. T-DM1PC vs. PTC_T-DM1P [11], while one trial comprised four arms that compared TC vs. PC vs. PT vs. PTC [38]. The duration of treatment ranged from 12 to 30 weeks with different dosages and cycle lengths (see Appendix A). The median age ranged from 48 to 53 years, and nodal positivity ranged from 20% to 80%. Tumor grade was largely T2 and T3, and at least 40% or more of patients were hormone receptor negative (i.e., both estrogen and progesterone receptors negative). All 21 RCTs reported pCR and SAE, while BCS, DFS, and overall survival were reported in 12, 10, and 7 RCTs, respectively. Only 7 and 4 RCTs provided KM curves for DFS and overall survival, respectively, with the latter insufficient for pooling.

### 3.1. Risk of Bias Assessment

The results of the risk of bias assessment for the pCR, BCS, DFS, and SAE outcomes are presented in Appendix A. The majority of the studies for pCR (13/21), BCS (9/12), and DFS (4/7) had a low risk of bias, and none had a high risk of bias. For the SAE outcome, only a single study had an overall high risk of bias due to missing outcome data, with 12 studies registering some concerns.

### 3.2. Pathological Complete Response

#### 3.2.1. Pairwise Meta-Analysis

The pathological complete response was reported in 21 studies; four treatment comparisons (i.e., TC vs. C, LC vs. TC, LTC vs. TC, and LTC vs. LC) were pooled, indicating significantly higher pCR in TC vs. C [29,33,34,35,45], LTC vs. TC [31,40,43,44,47,48,53], and LTC vs. LC [31,40,43,44,47,48,53], with corresponding pooled RRs (95% CI) of 1.81 (1.36, 2.42; I^2^ = 0%), 1.26 (1.11, 1.42; I^2^ = 1.8%), and 1.66 (1.33, 2.06; I^2^ = 42%) (see Appendix A). Conversely, LC showed significantly lower pCR than TC, with a pooled RR (95% CI) of 0.74 (0.63, 0.87); I^2^ = 26%) [31,39,40,43,44,46,47,48,53] (see Appendix A).

Given the high I^2^ value observed in the LTC vs. LC comparison, sources of heterogeneity were explored (see Appendix A). Accounting for types of chemotherapy that decreased the I^2^ value from 42% to 0%, a subgroup analysis showed that RR was higher with taxane only than taxane plus anthracycline, with pooled RRs (95% CI) of 2.13 (1.65, 2.75) and 1.33 (1.12, 1.58), respectively (see Appendix A). There was no significant variation in the comparisons from other subgroup analyses (Appendix A). For LC vs. TC (see Appendix A), subgroup analyses showed reductions in heterogeneity when accounting for age, percentages of nodal positive, and T3 and T4, although none reached significance (Appendix A).

Egger’s tests did not provide any evidence of publication bias for any pCR pooled estimates (see Appendix A), in support of the funnel plots, with the exception for the comparison between TC vs. C, which was asymmetrical (see Appendix A). A contour-enhanced funnel plot suggested the asymmetry may be a consequence of a small study effect (Appendix A).

#### 3.2.2. Network Meta-Analysis

The NMA for pCR included 21 studies with 16 comparisons (Figure 2A) and showed consistency (global test chi-square = 2.01, *p*-value = 0.571). The data used for pooling are described in Appendix A. All anti-HER2 regimens except PC, LC, and PT significantly increased pCR when compared to chemotherapy alone, with RRs (95% CI) of 4.03 (2.22, 7.30) for PTC_T-DM1P, 3.25 (1.80, 5.88) for T-DM1PC, 4.42 (2.25, 8.67) for T-DM1LC, 2.24 (1.58, 3.18) for LTC, 3.22 (2.04, 5.09) for PTC, 2.57 (1.48, 4.45) for T-DM1P, 3.08 (1.68, 5.62) for T-DM1, and 1.81 (1.32, 2.48) for TC (see Table 2). Both LC and PT regimens had significantly lower pCR when compared to TC. All dual anti-HER-2 agents plus chemotherapy (i.e., PTC, LTC, T-DM1LC, T-DM1PC, and PTC_T-DM1P) had significantly higher pCR than single anti-HER-2 agents plus chemotherapy regimens (i.e., LC, PC, TC) except LTC vs. PC, with pooled RRs (95% CI) of 1.78 (1.28, 2.48) for PTC vs. TC, 2.44 (1.34, 4.43) for T-DM1LC vs. TC, 2.22 (1.34, 3.68) for PTC_T-DM1P vs. TC, 1.79 (1.09, 2.97) for T-DM1PC vs. TC, and 1.23 (1.06, 1.44) for LTC vs. TC. When compared to PC, pooled RRs were 2.14 (1.27, 3.61) for PTC, 2.93 (1.42, 6.03) for T-DM1LC, 2.67 (1.40, 5.11) for PTC_T-DM1P, 2.16 (1.13, 4.12) for T-DM1PC, and, when compared to LC, pooled RRs were 2.41 (1.66, 3.48) for PTC, 1.67 (1.40, 1.99) for LTC, 3.30 (1.78, 6.13) for T-DM1LC, 2.43 (1.43, 4.12) for T-DM1PC, and 3.01 (1.77, 5.12) for PTC_T-DM1P (see Table 2). T-DM1 also had significantly higher pCR compared to LC, PC, and TC (see Table 2). In addition, all dual anti-HER-2 agents plus chemotherapy regimens T-DM1P and T-DM1 also significantly increased pCR when compared to PT (see Table 2). However, the chance of having pCR with dual anti-HER-2 agents plus chemotherapy regimens did not significantly differ from T-DM1P and T-DM1 regimens. In addition, both T-DM1LC and PTC_T-DM1P had significantly higher pCR compared to LTC, with RRs (95% CI) of 1.97 (1.06, 3.66) and 1.80 (1.06, 3.05), respectively. A SUCRA plot identified T-DM1LC with the highest probability of being the best regimen (91.9%), followed by PTC_T-DM1P (90.5%) and PTC (74.8%) (see Appendix A). Adjusted funnel plots were symmetrical, indicating no publication bias (see Appendix A).

### 3.3. Serious Adverse Events

Twenty-one studies reported grade 3 and 4 adverse events that were included in the analysis of SAE [10,31,32,35,40,45,46,47,48,53,54].

#### 3.3.1. Pairwise Meta-Analysis

Of the 21 studies, treatment comparisons included TC vs. C (five studies), LC vs. TC (nine studies), LTC vs. TC (seven studies), and LTC vs. LC (seven studies). The SAEs were not significantly different for TC vs. C and LTC vs. LC (see Appendix A), whereas LC and LTC had significantly higher SAEs than TC with pooled RRs (95% CI) of 1.43 (1.08, 1.89; I^2^ = 79.97%) and 1.77 (1.22, 2.59; I^2^ = 73.78%), respectively (Appendix A). 

The type of chemotherapy included could reduce the level of heterogeneity (Tau^2^) for both LC vs. TC and LTC vs. TC, respectively (Appendix A). The subgroup analyses indicated that risks of SAEs in LC and LTC were higher than TC for the taxane regimen alone compared to taxane plus anthracycline regimens (Appendix A).

There was no evidence of publication bias as indicated by either Egger’s tests or funnel plots for TC vs. C and LTC vs. LC (Appendix A). The funnel plots for LC vs. TC and LTC vs. TC were asymmetrical (Appendix A), which may be due to heterogeneity, as suggested from the contour-enhanced funnel plots (Appendix A).

#### 3.3.2. Network Meta-Analysis

Data used for NMA of SAE are described in Appendix A. A global test (chi-square = 7.87, *p*-value = 0.049) suggested evidence of inconsistency across the 21 studies considered. An inconsistency factor (IF) plot identified the TC-PT-PTC loop with the highest IF (1.90, 95% CI: 0.31, 3.48) (Appendix A). Study characteristics in this loop were explored (see Appendix A), and the trial by Nitz et al. [49] included only hormonal receptor negative breast cancers. After removing this study, the network was deemed consistent (chi-square = 3.24, *p*-value = 0.198) (see Figure 2B). Relative treatment effects were estimated, indicating that single and dual anti-HER2 regimens without chemotherapy (i.e., T-DM1, PT, T-DM1P) had significantly lower SAE risk compared to single anti-HER2 agents with chemotherapy (i.e., TC, PC, LC) (see Table 2). All dual anti-HER2 agents with chemotherapy (i.e., PTC, LTC, T-DM1LC, T-DM1PC, and PTC_T-DM1P) also had significantly higher SAE risk compared to dual anti-HER2 regimens without chemotherapy (i.e., PT and T-DM1P), with pooled RRs (95% CI) of 6.72 (2.06, 21.89) for T-DM1PC vs. PT, 12.14 (3.77,39.07) for PTC_T-DM1P vs. PT, and 2.48 (1.15,5.34) for T-DM1PC vs. T-DM1P, 4.48 (2.12,9.47) for PTC_T-DM1P vs. T-DM1P, respectively. The SAE risk in the T-DM1 regimen was not significantly different compared to dual anti-HER2 regimens without chemotherapy (i.e., PT and T-DM1P) (see Table 2). Among the regimens with chemotherapy, T-DM1PC had a significantly lower risk of SAE than TC, PC, LC, as well as PTC and LTC, with pooled RRs (95% CI) of 0.48 (0.26,0.91), 0.46 (0.23,0.94), 0.37 (0.19, 0.73), 0.50 (0.30, 0.85), and 0.34 (0.17, 0.69), respectively (see Table 2). In addition, the risk of SAE was significantly higher in LTC compared to TC, with a pooled RR of 1.40 (1.04, 1.89). According to SUCRA, PT had the highest probability of having the lowest risk of SAE (99.2%), followed by T-DM1P (88%), T-DM1 (83.9%), and T-DM1PC (71.6%); LTC was associated with the poorest risk of SAE (10.7%) (see Appendix A). The comparison-adjusted funnel plots were symmetrical, suggesting no evidence of publication bias (Appendix A).

### 3.4. Ranking of Regimens According to Efficacy and Safety 

A cluster rank plot was constructed considering the SUCRA of pCR on the *x*-axis and SUCRA of lowering SAEs on the *y*-axis (Figure 3). T-DM1LC was located in the bottom right of the graph, representing the highest pCR with high SAEs, while T-DM1PC, PTC, T-DM1, and PTC_T-DM1P still ranked high for pCR but had better SAE profiles. Therefore, T-DM1PC, T-DM1, PTC, and PTC_T-DM1P provided an optimal balance between efficacy and risk of SAE.

### 3.5. Breast Conservation Surgery

BCS was reported in 12 of the 21 studies [11,29,31,36,38,40,41,43,45,47,50,54].

#### 3.5.1. Pairwise Meta-Analysis

Four treatment comparisons (i.e., TC vs. C (three studies) [30,36,45], LC vs. TC (five studies) [31,41,43,46,47], LTC vs. TC (four studies) [31,41,43,47], and LTC vs. LC (four studies) [31,41,43,47]) had sufficient data for pooling. None of these studies showed significant differences of BCS between treatment comparisons, with corresponding pooled RRs (95% CI) of 1.10 (0.73, 1.64; I^2^ = 53.01%) for TC vs. C, 0.97 (0.83, 1.14; I^2^ = 25.34%) for LC vs. TC, 1.00 (0.87, 1.14; I^2^ = 0%) for LTC vs. TC, and 1.05 (0.91, 1.21; I^2^ = 0%) for LTC vs. LC (Appendix A). 

#### 3.5.2. Network Meta-Analysis

The NMA included 12 studies with 11 treatment regimens that passed the consistency assumption (global test chi-square = 0.16, *p*-value = 0.6937) (see Figure 2C). There were no significant differences in BCS for any of the treatment comparisons except for PTC vs. T-DM1P, with a pooled RR (95% CI) of 1.26 (1.02, 1.55) (see Appendix A). PTC (68.5%) was ranked best in terms of having the highest BCS (Appendix A). The comparison-adjusted funnel plots were symmetrical and indicated no evidence of publication bias (Appendix A).

### 3.6. Disease-Free Survival

Of the 21 RCTs, 10 reported DFS [5,11,12,30,33,37,42,52,54,56], but only seven studies [5,12,37,42,52,54,56] provided KM curves from which IPD could be simulated. The follow-up time ranged from 36 to 168 months, with a median of 48 months. The regimens included were C, TC, PC, LC, PT, PTC, LTC, T-DM1P, and T-DM1. The mixed-effect parametric survival analysis identified PTC with the highest DFS, in contrast to C, which had the lowest DFS (see Figure 4). Compared to TC, chemotherapy alone had a significantly higher risk of disease recurrence (HR = 1.70 95% CI: 1.15–2.51), while PTC had a significantly lower risk of disease recurrence with an HR (95% CI) of 0.54 (0.32–0.91). Disease recurrence in PC, LC, PT, LTC, T-DM1P, and T-DM1 did not differ significantly from TC (see Table 3).

## 4. Discussion

Our results suggest that T-DM1LC has the highest probability of achieving pCR, followed by PTC_T-DM1P and PTC, respectively. In addition, PTC also provided the longest DFS. PT was ranked first (i.e., lowest risk) for SAE, followed by T-DM1P and T-DM1. When considering both the optimal benefit (pCR or DFS) and risk (SAE) together, T-DM1PC, PTC_T-DM1P, T-DM1, or PTC were considered the best regimens for neoadjuvant anti-HER2 therapy.

Previous NMAs by Wu et al. [9] and Nakashoji et al. [8] identified PTC as the best neoadjuvant anti-HER2 regimen for achieving pCR in early-stage breast cancer. With our updated data, T-DM1LC now represents the top-ranked treatment, with PTC dropping to the third-ranked treatment in terms of pCR outcomes. In this fast-moving field of cancer therapeutics, the more recently available regimens of T-DM1 were not considered within the previous NMAs, including T-DM1, T-DM1LC, T-DM1PC, and PTC_T-DM1P. Trastuzumab emtansine consists of trastuzumab and the cytotoxic agent DM1 (derivative of maytansine), which can directly deliver cytotoxic molecules to tumors, potentially increasing the efficacy in comparison to trastuzumab [57,58]. Beyond pCR outcomes, our study also assessed the efficacy of neoadjuvant anti-HER2 therapies to prolong DFS, an outcome not considered in previous NMAs. PTC significantly increased DFS compared to TC, while none of the other anti-HER2 regimens significantly differed from TC. Nevertheless, there were some limitations associated with the more recent T-DM1 regimens, such as T-DM1 and T-DM1P, which had insufficient data to properly evaluate this outcome.

Although T-DM1LC was ranked the best for the pCR outcome, its SAE rate was very high. This may be due to lapatinib, which is associated with a higher risk of SAE grades 3–4, such as neutropenia, diarrhea, and hepatotoxicity [31,40,43,59]. Our study also indicated that all the regimens containing lapatinib (i.e., LTC, LC, and T-DM1LC) were ranked the worst for SAE. In addition to lapatinib, SAE grades 3–4 were commonly found in regimens with chemotherapy. Three regimens without chemotherapy (i.e., PT, T-DM1, and T-DM1P) were the best treatments for lowering SAEs. However, the efficacy of PT in terms of pCR and DFS was very low in comparison to the other regimens.

Moreover, the percentage of patients who discontinued the treatment was high in regimens with lapatinib. However, most studies applied intention to treat analysis for analyzing the data. Although this analysis might underestimate the true treatment efficacy in the ideal situation, it usually reflects the efficacy of the drugs when using them in the real-clinical setting where some patients may not be complying well with the treatment regimen.

When considering the benefit (pCR) and risk (SAE) together, T-DM1PC and T-DM1 were identified as the optimal neoadjuvant therapy for early-stage HER2+ve breast cancer given the greater chance of achieving pCR and the low risk of SAE. PTC_T-DM1P was identified as the next-best alternative regimen given the higher pCR to T-DM1PC coupled with the greater risk of SAE compared to T-DM1PC. These conclusions contrast to those from previous NMAs, which indicated PTC as the most effective treatment. However, our NMA has been updated to include the most recent treatment regimens and confirms that switching trastuzumab to T-DM1 in combination with PC could offer increased efficacy coupled with decreased risk of SAE. In addition, our findings failed to support chemotherapy alone or single anti-HER2 regimens in combination with chemotherapy (i.e., TC, PC, LC) as optimal neoadjuvant therapies for the treatment of early-stage HER2+ve breast cancer due to a very low chance of achieving pCR and an associated increased risk of SAE.

Although the recommended regimens of T-DM1 were top-ranked in our cluster ranking plot, its prohibitively high cost of $127,035 USD per patient per year should be considered [60]. Therefore, affordability and accessibility may be issues, especially in low- and middle-income countries, warranting economic evaluation before further consideration of these regimens.

### Strengths and Limitations

To our knowledge, our NMA is the most up to date, and includes novel neoadjuvant anti-HER2 regimens with combinations of T-DM1 and chemotherapy, providing the most current evidence for treatment recommendations for HER2+ve breast cancer. Moreover, DFS is a clinically important outcome measure not considered in previous NMAs. However, not all novel regimens provided sufficient DFS outcome data for pooling, especially those regimens that included T-DM1. Therefore, we were unable to compare the regimen efficacy for DFS outcomes between new neoadjuvant anti-HER2 regimens, including T-DM1 and PTC, with other regimens, and, as such, recommendations on the optimal treatment regimens were mainly based on the outcomes of pCR and SAE. Nevertheless, several RCTs and meta-analyses suggest pCR following neoadjuvant therapy in HER2+ve breast cancer is a valid surrogate of long-term outcomes [61,62,63].

Our study also had some limitations. First, there were a small number of studies and participants that included newer treatment regimens, such as T-DM1 (i.e., T-DM1PC, T-DM1LC, PTC_T-DM1P, and T-DM1). We pooled the treatment effects based on a small number of included studies, each of which also had small sample sizes, which might result in the imprecision or uncertainty of treatment effects estimated from the network meta-analysis. Therefore, additional RCTs are needed to substantiate the findings from our study. Second, due to insufficient data, we were unable to analyze overall survival outcomes, which is the most important outcome of cancer treatment. Nevertheless, it has been shown that DFS is an adequate proxy for overall survival in HER2+ve breast cancer patients [64]. Third, all the included studies were funded by the pharmaceutical companies, which might have an influence on conducting the study, data analysis, or reporting the results. Therefore, further updated pooling treatment effects are required when there are more studies with new treatment regimens and/or with non-profit sponsors.

## 5. Conclusions

In conclusion, the T-DM1PC, T-DM1, and PTC_T-DM1P regimens had the optimal balance between efficacy (pCR) and safety (SAE) compared to other neoadjuvant anti-HER2 regimens for early-stage and locally advanced HER2+ve breast cancer, while DFS was the highest for the PTC regimen. Nonetheless, the results of regimens with T-DM1 are based on a small number of studies. Thus, additional RCTs to assess the efficacy of neoadjuvant regimens with T-DM1 are still needed to confirm these findings.

## Figures and Tables

**Figure 1 cancers-14-00523-f001:**
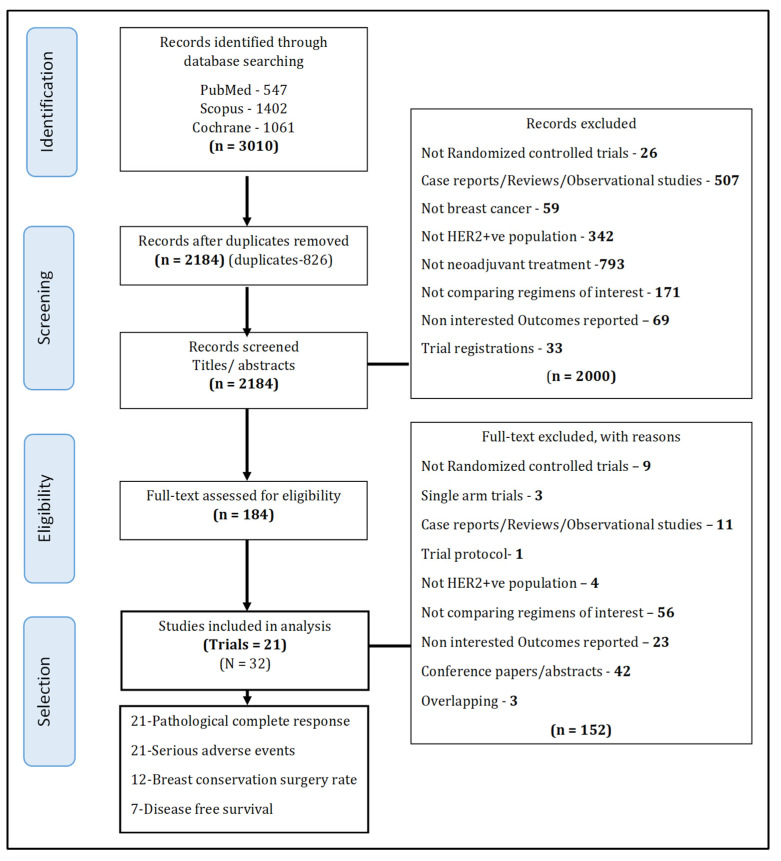
Preferred Reporting Items for Systematic Reviews and Meta-Analyses (PRISMA) flow diagram.

**Figure 2 cancers-14-00523-f002:**
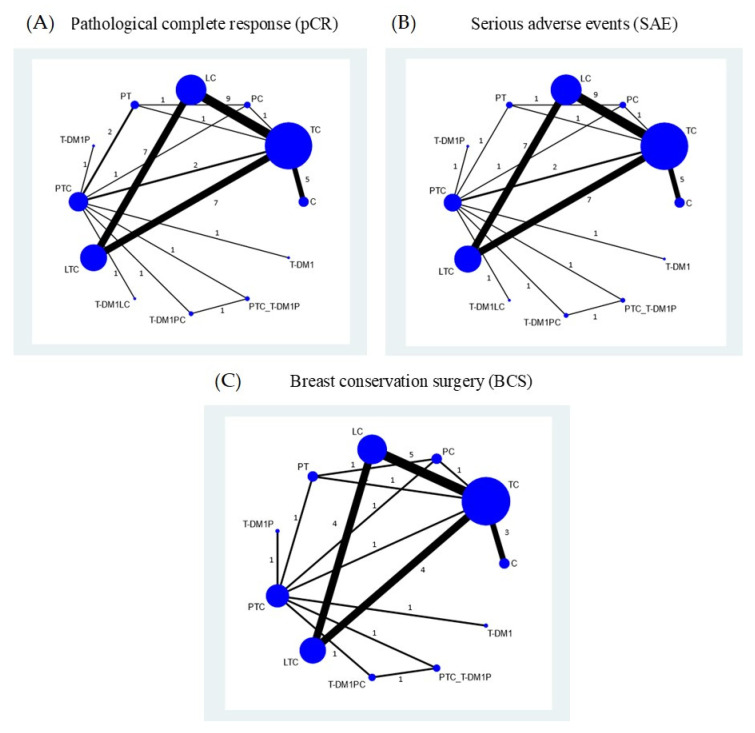
Network maps for (**A**) pathological complete response (pCR), (**B**) serious adverse events (SAE), and (**C**) breast conservation surgery (BCS)**.** Nodes and lines represent interventions and direct treatment comparisons with weighted size according to the number of studies for each direct comparison. The numbers on the lines indicate the number of studies for each treatment comparison. C = chemotherapy; TC = Trastuzumab + chemotherapy; PC = Pertuzumab + chemotherapy; LC = Lapatinib + chemotherapy; PTC = Pertuzumab + trastuzumab + chemotherapy; LTC = Lapatinib + trastuzumab +chemotherapy; PT = Pertuzumab + trastuzumab; T-DM1P = trastuzumab emtansine+ pertuzumab; T-DM1LC = trastuzumab emtansine + lapatinib + chemotherapy; T-DM1PC = trastuzumab emtansine + pertuzumab + chemotherapy; PTC_T-DM1P = pertuzumab + trastuzumab + chemotherapy followed by trastuzumab emtansine + pertuzumab; T-DM1 = Trastuzumab emtansine.

**Figure 3 cancers-14-00523-f003:**
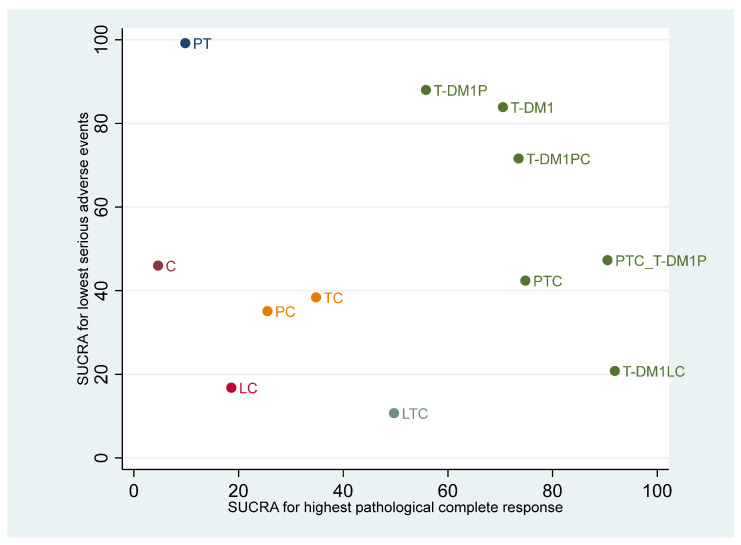
Clustered ranking plot of surface under the cumulative ranking curves (SUCRAs) for lowest probability of serious adverse events versus SUCRAs for highest probability of pathological complete response outcomes. C = chemotherapy; TC = Trastuzumab + chemotherapy; PC = Pertuzumab + chemotherapy; LC = Lapatinib + chemotherapy; PTC = Pertuzumab + trastuzumab + chemotherapy; LTC = Lapatinib + trastuzumab +chemotherapy; PT = Pertuzumab + trastuzumab; T-DM1P = trastuzumab emtansine+ pertuzumab; T-DM1LC = trastuzumab emtansine + lapatinib + chemotherapy; T-DM1PC = trastuzumab emtansine + pertuzumab + chemotherapy; PTC_T-DM1P = pertuzumab + trastuzumab + chemotherapy followed by trastuzumab emtansine + pertuzumab; T-DM1 = Trastuzumab emtansine.

**Figure 4 cancers-14-00523-f004:**
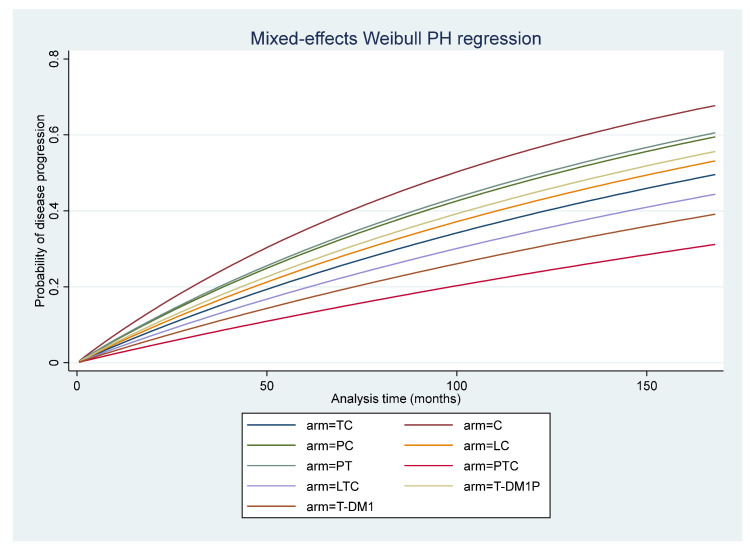
Mixed-effect parametric survival analysis of disease-free survival. C = chemotherapy; TC = Trastuzumab + chemotherapy; PC = Pertuzumab + chemotherapy; LC = Lapatinib + chemotherapy; PTC = Pertuzumab + trastuzumab + chemotherapy; LTC = Lapatinib + trastuzumab +chemotherapy; PT = Pertuzumab + trastuzumab; T-DM1P = trastuzumab emtansine+ pertuzumab; T-DM1 = Trastuzumab emtansine.

**Table 1 cancers-14-00523-t001:** Characteristics of the included randomized controlled trials.

Study	Country	Total Participants	Regimen	No. Per Arm	Age (Yrs) Median (Range)	Stage of Breast Cancer			Median Size of Tumor (mm)	T Stage				% Nodal Positive	% Hormone Receptor Negative	%Pre-menopausal	% Withdrawal	% Discontinuation of Treatment	Funding Source
						I	II	III		T1	T2	T3	T4						
Buzdar et al., 2005, Buzdar et al., 2007 [29,30]	USA	42	C	19	48 (25–75)	-	-	-	-	2	13	4	0	63	42	-	NR	NR	Genentech, Pfizer, and Bristol Myers Squibb
			TC	23	52 (29–71)	-	-	-	-	2	15	5	1	57	43	-	NR	NR	
CHERLOB (2012, 2021) [31,56]	Italy	121	TC	36	50 (34–65)	-	30	6	30	-	-	-	-	-	42	47	0 (0%)	0 (0%)	GSK
			LC	39	49 (34–68)		32	23	30	-	-	-	-	-	38	39	1 (2.56%)	12 (30.77%)	
			LTC	46	49 (26–65)		13	9	35	-	-	-	-	-	39	42	1 (2.17%)	8 (17.39%)	
REMAGUS 02 (2010, 2017) [32,33]	France	120	C	58	-	-	-	-	-	-	27	31 (T3/T4)		67	36	-	0 (0%)	0 (0%)	French Programme Hospitalier de Recherche Clinique, Pfizer Inc. France, Roche Pharmaceutical, and Sanofi-Aventis.
			TC	62	-	-	-	-	-	-	32	30 (T3/T4)		60	45	-	0 (0%)	5 (8.06%)	
Chang et al., 2010 [34]	USA	30	TC	15	-	-	-	-	-	-	-	-	-	-	-	-	NR	NR	RO1, Aventis, Genentech
			C	15		-	-	-	-	-	-	-	-	-	-	-	NR	NR	
NOAH (2010, 2011, 2014) [35,36,37]	International	235	TC	117	-	-	-	-	-	-	-	-	49	85	-	-	0 (0%)	4 (3.42%)	F Hoff mann-La Roche
			C	118	-	-	-	-	-	-	-	-	51	84	-	-	0 (0%)	9 (7.63%)	
NeoSphere (2012, 2016) [5,38]	International	417	TC	107	50 (32–74)	-	-	-	50	-	-	-	-	70	53	-	4 (3.74%)	0 (0%)	F Hoff mann-La Roche
			PTC	107	50 (28–77)	-	-	-	55	-	-	-	-	70	53	-	5 (4.67%)	1 (0.93%)	
			PT	107	49 (22–80)	-	-	-	50	-	-	-	-	70	52	-	14 (13.08%)	0 (0%)	
			PC	96	49 (27–70)	-	-	-	50	-	-	-	-	71	52	-	6 (6.25%)	2 (2.08%)	
GeparQuinto, GBG 44 (2012, 2018) [12,39]	Germany	615	TC	307	50 (25–74)	5	159	141	-	51	180	17	-	68	45	-	0 (0%)	32 (10.42%)	GSK, Roche, and Sanofi -Aventis
			LC	308	50 (21–73)	3	141	152	-	45	184	18	-	68	44	-	0 (0%)	51 (16.56%)	
NeoALTTO (2012, 2013, 2019) [40,41,42]	International	455	LC	154	50 (42–56)	-	-	-	-	-	-	-	-		49	-	5 (3.25%)	47 (30.52%)	GSK
			TC	149	49 (44–57)	-	-	-	-	-	-	-	-		51	-	2 (1.34%)	10 (6.71%)	
			LTC	152	50 (43–59)	-	-	-	-	-	-	-	-		51	-	4 (2.63%)	55 (36.18%)	
NSABP protocol B-41 (2013) [43]	USA	529	TC	181	-	-	-	-	-	-	102	79 (T3/T4)		51	31	-	2 (1.10%)	40 (23%)	GSK
			LC	174	-	-	-	-	-	-	81	93 (T3/T4)		52	40	-	3 (1.72%)	55 (35%)	
			LTC	174	-	-	-	-	-	-	88	86 (T3/T4)		49	37	-	2 (1.15%)	61 (37%)	
LPT109096 (2013) [44]	USA	100	TC	33	51.1 (21–67)	-	-	-	-	-	22	8	3	45	-	42	0 (0%)	7 (21.21%)	GSK
			LC	34	52 (25–67)	-	-	-	-	-	12	11	8	68	-	44	0 (0%)	5 (14.71%)	
			LTC	33	49.2 (28–66)	-	-	-	-	-	22	6	5	61	-	52	5 (15.15%)	10 (30.30%)	
ABCSG-24 (2014) [45]	Austria	93	C	49	48 (29–68)	-	-	-	-	9	24	15	1	51	39	57	NR	NR	Amgen Austria, Roche Austria, SanofiAventis Austria, and EBEWE Austria
			TC	44	50 (26–70)	-	-	-	-	8	25	1	3	55	41	57	NR	NR	
GEICAM/ 2006-14 (2014) [46]	Spain	102	TC	50	48.5 (32–74)	-	-	-	33	6	31	4	9	74	21	58	0 (0%)	2 (4.0%)	GSK
			LC	52	48 (30–79)	-	-	-	35	8	29	8	7	63	22	54	0 (0%)	10 (19.23%)	
EORTC 10054 (2015) [47]	France, Europe	128	LC	23	49.9 (27.3–68.5)	-	-	-	-	1	11	8	3	70	36	-	1 (4.35%)	4.5%	US National Cancer Institute, Fonds Cancer (FOCA) Belgium; Cancer Research UK and French Ligue Nationale Contre le Cancer, GSK
			TC	53	47 (25.3–68.9)	-	-	-	-	0	24	19	10	66	48	-	1 (1.89%)	9.4%	
			LTC	52	(27.3–70.8)	-	-	-	-	0	28	13	9	63	52	-	4 (7.69%)	26%	
CALGB 40601 (2016, 2020) [48,55]	USA	299	LTC	117	48 (24–70)	-	80	37	40	-	-	-	-	-	40	62	1 (0.85%)	26 (22.22%)	National Cancer Institute (NCI), GSK, and the University of North Carolina
			TC	118	50 (30–75)	-	80	38	40	-	-	-	-	-	41	53	1 (0.85%)	10 (8.47%)	
			LC	64	50 (25–74)	-	47	17	40	-	-	-	-	-	44	56	2 (3.13%)	23 (35.94%)	
WSG-ADAPT (2017) [49]	Germany	134	PT	92	54	-	-	-	-	38	47	6	1	46	100	-	0 (0%)	8 (8.70%)	Hoffmann la Roche
			PTC	42	51.5	-	-	-	-	17	22	3	0	38	100	-	0 (0%)	4 (9.52%)	
KRISTINE (2018, 2019) [50,52]	International	444	T-DM1P	223	50 (42–57)	-	-	-	-	-	-	-	-	-	38	-	0 (0%)	16 (7.17%)	Hoffmann-La Roche and Genentech
			PTC	221	49 (41–57)	-	-	-	-	-	-	-	-	-	38	-	0 (0%)	8 (3.62%)	
Teal study (2019) [10]	USA	30	T-DM1LC	14	53.1 (29–70)		8	6	-	-	-	-	-	-	-	-	NR	NR	Celgene and Novartis
			PTC	16	57.2 (40–75)		7	9	-	-	-	-	-	-	-	-	NR	NR	
PEONY (2019) [51]	China	329	PTC	219	49 (24–72)	-	-	-	-	-	155	45	19	27	48	60	1 (0.46%)	4 (1.83%)	Hoffmann-La Roche
			TC	110	49 (27–70)	-	-	-	-	-	71	29	10	19	49	59	0 (0%)	2 (1.82%)	
Masuda et al., 2020 [11]	Japan	204	PTC	51	53 (28–70)				27.0	11	37	3		33	41	55	0 (0%)	2 (3.92%)	Japan Breast Cancer Research Group (JBCRG). Chugai Pharmaceutical Co., Ltd.
			PTC_ T-DM1P	52	53 (29–69)	-	-	-	25.5	13	35	4	-	40	44	56	0	3 (5.77%)	
			T-DM1PC	101	52 (25–70)	-	-	-	27	20	72	9	-	37	42	53	0 (0%)	5 (4.95%)	
TRIO-US B07 (2020) [53]	USA	128	TC	34	48	2	20	12	55.4	-	-	-	-	-	41	-	0 (0%)	0 (0%)	Sanofi aventis and GSK
			LC	36	51	1	28	7	51.6	-	-	-	-	-	50	-	0 (0%)	10 (27.78%)	
			LTC	58	47	3	38	17	41.5	-	-	-	-	-	41	-	0 (0%)	15 (25.86%)	
Hatschek et al. (2021) [54]	Sweden	198	PTC	99	51	-	-	-	-	14	63	17	-	61.6	33	50.5	0 (0%)	17 (17.17%)	Region Stockholm, Karolinska Institutet, Swedish Research Council, Swedish Cancer Society, Roche Sweden
			T-DM1	99	53	-	-	-	-	20	61	17	-	50.5	40	46.8	1 (1.01%)	9 (9.09%)	

C = Chemotherapy; TC = Trastuzumab + chemotherapy; PC = Pertuzumab + chemotherapy; LC = Lapatinib + chemo-therapy; PTC = Pertuzumab + trastuzumab + chemotherapy; LTC = Lapatinib + trastuzumab +chemotherapy; PT = Pertuzumab + trastuzumab; T-DM1 = trastuzumab emtansine; T-DM1P = trastuzumab emtansine + pertuzumab; T-DM1LC = trastuzumab emtansine +lapatinib + chemotherapy; T-DM1PC = trastuzumab emtansine + pertuzumab + chemotherapy; PTC_T-DM1P = pertuzumab + trastuzumab + chemotherapy followed by trastuzumab emtansine + pertuzumab.

**Table 2 cancers-14-00523-t002:** Risk ratios and 95% confidence intervals of network meta-analysis of pathological complete response (above grey diagonal line) and serious adverse drug events (below grey diagonal line).

Risk Ratio (95% Confidence Intervals)											
**C**	**1.81 (1.32, 2.48)**	1.51 (0.79, 2.87)	1.34 (0.94, 1.91)	1.11 (0.65, 1.92)	**2.57 (1.48, 4.45)**	**3.22 (2.04, 5.09)**	**2.24 (1.58, 3.18)**	**4.42 (2.25, 8.67)**	**3.25 (1.80, 5.88)**	**4.03 (2.22, 7.30)**	**3.08 (1.68, 5.62)**
1.08 (0.79,1.48)	**TC**	0.83 (0.47, 1.46)	**0.74 (0.63, 0.87)**	**0.61 (0.39, 0.96)**	1.42 (0.90, 2.22)	**1.78 (1.28, 2.48)**	**1.23 (1.06, 1.44)**	**2.44 (1.34, 4.43)**	**1.79 (1.09, 2.97)**	**2.22 (1.34, 3.68)**	**1.70 (1.02, 2.84)**
1.13 (0.64, 2.00)	1.04 (0.64, 1.68)	**PC**	0.89 (0.49, 1.59)	0.74 (0.41, 1.33)	1.70 (0.93, 3.12)	**2.14 (1.27, 3.61)**	1.48 (0.83, 2.66)	**2.93 (1.42, 6.03)**	**2.16 (1.13, 4.12)**	**2.67 (1.40, 5.11)**	**2.04 (1.06, 3.93)**
1.42 (0.94, 2.14)	**1.31 (1.01, 1.70)**	1.26 (0.73, 2.17)	**LC**	0.83 (0.52, 1.33)	**1.92 (1.19, 3.10)**	**2.41 (1.66, 3.48)**	**1.67 (1.40, 1.99)**	**3.30 (1.78, 6.13)**	**2.43 (1.43, 4.12)**	**3.01 (1.77, 5.12)**	**2.30 (1.34, 3.94)**
**0.08 (0.03, 0.23)**	**0.07 (0.02, 0.21)**	**0.07 (0.02, 0.20)**	**0.05 (0.02, 0.16)**	**PT**	**2.31 (1.47, 3.61)**	**2.89 (2.08, 4.01)**	**2.01 (1.26, 3.21)**	**3.96 (2.18, 7.19)**	**2.92 (1.77, 4.82)**	**3.62 (2.19, 5.98)**	**2.76 (1.65, 4.61)**
**0.21 (0.10, 0.44)**	**0.19 (0.10, 0.38)**	**0.19 (0.09, 0.40)**	**0.15 (0.07, 0.31)**	2.71 (0.81, 9.06)	**TDM1P**	1.25 (0.92, 1.70)	0.87 (0.54, 1.40)	1.72 (0.96, 3.08)	1.27 (0.78, 2.06)	1.57 (0.96, 2.55)	1.20 (0.73, 1.97)
1.04 (0.64, 1.68)	0.96 (0.66, 1.38)	0.92 (0.56, 1.50)	0.73 (0.46, 1.15)	**13.31 (4.60, 38.52)**	**4.91 (2.77, 8.68)**	**PTC**	0.69 (0.48, 1.00)	1.37 (0.83, 2.25)	1.01 (0.69, 1.47)	1.25 (0.86, 1.83)	0.96 (0.64, 1.42)
1.52 (0.99, 2.35)	**1.40 (1.04, 1.89)**	1.35 (0.77, 2.37)	1.07 (0.85, 1.35)	**19.55 (6.52, 58.63)**	**7.21 (3.44, 15.10)**	1.47 (0.92, 2.35)	**LTC**	**1.97 (1.06, 3.66)**	1.45 (0.86, 2.46)	**1.80 (1.06, 3.05)**	1.38 (0.80, 2.35)
2.37 (0.22, 25.64)	2.18 (0.21, 23.19)	2.10 (0.19, 22.78)	1.67 (0.15, 17.99)	**30.41 (2.34, 395.23)**	**11.21 (1.01, 123.92)**	2.29 (0.22, 23.58)	1.56 (0.14, 16.83)	**T-DM1LC**	0.74 (0.39, 1.38)	0.91 (0.49, 1.71)	0.70 (0.37, 1.31)
0.52 (0.26, 1.06)	**0.48 (0.26, 0.91)**	**0.46 (0.23, 0.94)**	**0.37 (0.19, 0.73)**	**6.72 (2.06, 21.89)**	**2.48 (1.15, 5.34)**	**0.50 (0.30, 0.85)**	**0.34 (0.17, 0.69)**	0.22 (0.02, 2.41)	**T-DM1PC**	1.24 (0.88, 1.74)	0.95 (0.55, 1.63)
0.94 (0.48, 1.87)	0.87 (0.47, 1.60)	0.84 (0.42, 1.67)	0.67 (0.34, 1.29)	**12.14 (3.77, 39.07)**	**4.48 (2.12, 9.47)**	0.91 (0.56, 1.48)	0.62 (0.32, 1.22)	0.40 (0.04, 4.33)	**1.81 (1.07, 3.05)**	**PTC_T-DM1P**	0.76 (0.44, 1.32)
**0.27 (0.11, 0.66)**	**0.25 (0.10, 0.58)**	**0.24 (0.09, 0.59)**	**0.19 (0.08, 0.46)**	3.41 (0.91, 12.74)	1.26 (0.48, 3.30)	**0.26 (0.12, 0.56)**	**0.17 (0.07, 0.43)**	0.11 (0.01, 1.31)	0.51 (0.20, 1.29)	**0.28 (0.11, 0.70)**	**T-DM1**

Results of treatment comparisons are read from right to left. For example, the risk ratio (95% confidence intervals) for pCR of TC vs. C is 1.81 (1.32, 2.48) and the risk ratio (95% confidence intervals) for SAE of TC vs. C is 1.08 (0.79, 1.48). Bold font indicates significance. C = Chemotherapy; TC = Trastuzumab + chemotherapy; PC = Pertuzumab + chemotherapy; LC = Lapatinib + chemotherapy; PTC = Pertuzumab + trastuzumab + chemotherapy; LTC = Lapatinib + trastuzumab +chemotherapy; PT = Pertuzumab + trastuzumab; T-DM1P = trastuzumab emtansine + pertuzumab; T-DM1LC = trastuzumab emtansine +lapatinib + chemotherapy; T-DM1PC = trastuzumab emtansine + pertuzumab + chemotherapy; PTC_T-DM1P = pertuzumab + trastuzumab + chemotherapy followed by trastuzumab emtansine + pertuzumab; T-DM1 = Trastuzumab emtansine.

**Table 3 cancers-14-00523-t003:** Hazard ratios of disease-free survival of neoadjuvant regimens.

Treatment Regimen	Predicted Median DFS (Months)	Hazard Ratio	95% Confidence Interval	*p*-Value
TC	173.23	1	-	-
C	100.92	1.70	1.15–2.51	0.008
PC	128.67	1.34	0.78–2.29	0.291
LC	155.27	1.11	0.87–1.43	0.403
PT	124.69	1.38	0.82–2.33	0.228
PTC	327.84	0.54	0.32–0.91	0.02
LTC	203.96	0.85	0.60–1.22	0.378
T-DM1P	144.18	1.20	0.61–2.33	0.598
T-DM1	243.35	0.72	0.30–1.72	0.457

C = Chemotherapy; TC = Trastuzumab + chemotherapy; PC = Pertuzumab + chemotherapy; LC = Lapatinib + chemotherapy; PTC = Pertuzumab + trastuzumab + chemotherapy; LTC = Lapatinib + trastuzumab +chemotherapy; PT = Pertuzumab + trastuzumab; T-DM1P = trastuzumab emtansine + pertuzumab; T-DM1 = Trastuzumab emtansine.

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
