# Peer review of "Neoadjuvant Treatment with HER2-Targeted Therapies in HER2-Positive Breast Cancer: A Systematic Review and Network Meta-Analysis"

_cancers, 2022, doi:10.3390/cancers14030523_

Round 1

Reviewer 1 Report

Thanks to the authors. Overall the paper reads well. It is difficult to determine the applicability of the findings however. In common with other meta-analyses, data were collected and analyzed basis of results reported from trials, instead of individual patient data. it is important the author are cautious in not overstating the findings. The ranking is an estimate of tolerability and effectiveness but further head to head studies are needed.

Reviewer 2 Report

NA

This manuscript is a resubmission of an earlier submission. The following is a list of the peer review reports and author responses from that submission.

Round 1

Reviewer 1 Report

Gunasekara et al. conducted a systematic review aimed to identify neoadjuvant anti-HER2 therapies with the best balance between efficacy and safety. They concluded that the T-DM1PC and PTC_T-DM1P regimens had the best balance between efficacy and safety, while DFS was highest for the PTC regimen.

The methods and results were clearly presented. However, the results and conclusion are similar to previous studies. In addition the results of T-DM1LC, based on a small study with 30 patients have been overemphasized.  

Reviewer 2 Report

the authors may reply following questions in section of discussion.

1. What is the impact on the result of NMA from the sample size of individual studies or phase II vs. phase III trials?

2. As the authors said ‘In this fast moving field of cancer therapeutics, …’ , if the manuscript accepted for publication and when being published, the current treatment guideline has been updating. What is the clinical implication of the results for daily practice?

Reviewer 3 Report

Thank you for the opportunity to review the article ‘Neoadjuvant Treatment with HER2 Targeted Therapies in HER2 Positive Breast Cancer: A Systematic Review and Network Meta-Analysis’. Overall, I thought it was a well constructed paper with sound methodology.

Please see some points for consideration below.

Methods

Did the authors consider searching ClinicalTrials.gov and contacting pharmaceutical companies marketing these medicines to ask for supplemental unpublished information about both premarketing and post-marketing studies.

Please describe who carried out the risk of bias assessments?

Discussion

Was length of follow up considered? Also was drop out rate looked at and cross over design? Participants who dropped out earlier tend to have poorer responses than those who remain on treatment, which are carried forward to the end of the trial by the LOCF analysis. The final result can be an underestimate of the true efficacy of the active drug.

Another possible explanation could be a bias in conduct, analysis, or reporting of head-to-head trials, driven by commercial interests. Was funding by industry captured anywhere. Interesting to consider  whether it is associated with substantial differences in terms of response or dropout rates. Perhaps this fits with Risk of Bias results.

Overall, it appears that drugs tended to show a greater effectiveness profiles when they were novel and used as experimental treatments than when they had become the older comparator drug, which fits with quick advances in field of oncology. I suggest emphasising this as primary finding.
